# Harmonizing across datasets to improve the transferability of drug combination prediction

Hanrui Zhang [1,6], Ziyan Wang [2,6], Yiyang Nan [1,6], Bulat Zagidullin [3,4,6], Daiyao Yi[1], Jing Tang [3✉] & Yuanfang Guan [1,5✉]

Combination treatment has multiple advantages over traditional monotherapy in clinics, thus becoming a target of interest for many high-throughput screening (HTS) studies, which enables the development of machine learning models predicting the response of new drug combinations. However, most existing models have been tested only within a single study, and these models cannot generalize across different datasets due to significantly variable experimental settings. Here, we thoroughly assessed the transferability issue of single-study-derived models on new datasets. More importantly, we propose a method to overcome the experimental variability by harmonizing dose–response curves of different studies. Our method improves the prediction performance of machine learning models by 184% and 1367% compared to the baseline models in intra-study and inter-study predictions, respectively, and shows consistent improvement in multiple cross-validation settings. Our study addresses the crucial question of the transferability in drug combination predictions, which is fundamental for such models to be extrapolated to new drug combination discovery and clinical applications that are *de facto* different datasets.

---

[1] Department of Computational Medicine and Bioinformatics, Michigan Medicine, University of Michigan, Ann Arbor, MI, USA. [2] Department of Electrical Engineering and Computer Science (EECS) - CSE Division, University of Michigan, Ann Arbor, MI, USA. [3] Research Program in Systems Oncology, Faculty of Medicine, University of Helsinki, Helsinki, Finland. [4] Institute for Molecular Medicine Finland (FIMM), University of Helsinki, Helsinki, Finland. [5] Department of Internal medicine, Michigan Medicine, University of Michigan, Ann Arbor, MI, USA. [6]These authors contributed equally: Hanrui Zhang, Ziyan Wang, Yiyang Nan, Bulat Zagidullin. ✉email: jing.tang@helsinki.fi; gyuanfan@umich.edu

Combining multiple therapeutic agents has become an emerging strategy in cancer treatment. While the monotherapy approach is often standard of care, the combination of multiple treatments has become inevitable as multiple comorbid conditions occur in cancer patients[1,2]. Moreover, drug combinations have shown advantages over monotherapy by overcoming drug resistance, and increasing efficacy through synergistic interactions[3]. To accelerate the development of new combination therapies, a large number of studies on high-throughput screening of drug combinations have been launched[4–6], and thereafter have been made comparable in large-scale databases such as DrugComb[7,8], DrugCombDB[9], and SYNERGxDB[10]. These databases provide abundant resources for training a powerful model to predict new potent combination treatments. For example, multiple machine learning tools have been developed, by hundreds of international participants in the NCI-DREAM Drug Sensitivity and Drug Synergy Challenge, and the AstraZeneca-Sanger Drug Combination Prediction (AZ-DREAM) Challenge[11,12].

However, most existing drug combination prediction models have been trained and tested using the same datasets[13–19]. Cross-dataset prediction still remains a significant challenge due to experimental variability between independent studies[20,21]. For example, when determining the drugs' efficacy, different dosing regimens are used. The O'Neil study used $5 \times 5$ dose–response matrices to determine the drug combination response[4], while the ALMANAC drug combinations were tested by $4 \times 4$ or $6 \times 4$ dose–response matrices[5]. While different dosages may not have a huge impact on summary monotherapy measurements, such as Hill coefficient (slope of the dose–response curve), $IC_{50}$ (dose at 50% of maximum response), $GR_{AoC}$ (area over the dose–response curve), and RI (relative inhibition normalized by positive control)[22,23], they may easily result in different interpolations of the dose–response curves, thus are often not used as features by machine learning models for cross-study drug combination prediction[24].

Due to the above challenges from different experimental settings, previous drug combination machine learning models only considered the summary monotherapy measurements as their dose–response features[11,14]. The complete dose–response curves of monotherapies, which contain the full spectrum of pharmacodynamics under different doses, cannot be fully captured by a single summary metric[25,26]. Therefore, a method for harmonizing different dose settings is crucial for cross-study drug combination machine learning models.

In this study, we propose to explore drug combination prediction across different studies with variable dose settings. In particular, we develop a method to standardize the dose–response curves across different studies. We show that such a method enables more efficient utilization of pharmacodynamics profiles of monotherapies in machine learning models, hence improving the prediction accuracy when transferring to new datasets. Our modeling strategy is of particular importance to solve the replicability issue of machine learning for drug combination discovery.

## Results

### A framework of intra- and inter-study machine learning prediction.
Our goal is to test the capability of machine learning models in predicting combination treatment response, not only within a single study but also between different studies and on unseen drug combinations. To achieve this goal, we first explore the publicly available high-throughput screening datasets for anticancer combination treatments, to build a gold standard for our experiment. We explore the current latest version of the DrugComb portal (https://drugcomb.org/), which contains the most comprehensive publicly available drug combination high-throughput screening datasets, including 24 independent studies[7,8]. Among them, we select four major datasets: ALMANAC, O'Neil, FORCINA, and Mathews, as they are of the biggest sizes and therefore are commonly used in machine learning prediction of combination responses[13,18,19,24,27–29]. These four studies contain a total of 406,479 drug combination experiments, 9,163 drugs, and 92 cell lines, while the size, drug, and cell line composition, as well as experimental settings, vary significantly among them (Supplementary Table 1). Of the four datasets, ALMANAC is the largest dataset with the most drug-cell line combinations, and FORCINA has the largest number of drugs screened. O'Neil has the best quality, where all the combinations are tested with four replicates, whereas ALMANAC tested at most three replicates for each combination and Mathews tested two replicates for each combination. In contrast, the FORCINA dataset contains no replicates.

We carried out a two-step cross-study validation (Fig. 1a). First, we trained dataset-specific models and carried out intra-study cross-validation. The cross-validation was set up so that the training and testing sets in this step do not share the same treatment-cell line combinations. Therefore, we aimed to test the performance of machine learning models in predicting unseen combination treatments within the same study. Next, during the inter-study cross-validation step, we tested these dataset-specific models on new individual datasets, which are denoted as "1 vs 1" in Fig. 1a. Furthermore, to explore more versatile cross-study scenarios, we designed a "3 vs 1" cross-validation strategy by combining three of the four datasets as the training set and the remaining one as the test set.

To analyze the potential of transferability, we determine the overlap of the drugs, cancer cell lines, and treatment-cell line combinations between the four studies (Fig. 1b). While drugs are overlapped between all the studies, no overlap of cell lines exists between FORCINA and Mathews with the other datasets, since both FORCINA and Mathews include only one unique cancer cell line. Overall, only 612 treatment-cell line combinations exist between ALMANAC and O'Neil, providing reference data for evaluating the performance of cross-dataset prediction.

Using the replicates within each dataset and the overlapping treatment-cell line combinations between the datasets, we analyzed the reproducibility of a drug combination sensitivity score called CSS[23], as well as multiple drug combination synergy scores, including S, Bliss, HSA, Loewe, and ZIP[23,30]. The intra- and inter-study reproducibility can be used as a benchmark for the drug combination prediction model we built in the next step (Supplementary Fig. 1). While no replicates existed in the FORCINA dataset, the O'Neil dataset showed the best intra-study replicability (0.93 Pearson's r for CSS, 0.929 Pearson's r for S, 0.778 for Bliss, 0.777 for HSA, 0.938 for Loewe, and 0.752 for ZIP), possibly due to the relatively more abundant replicates in this study. When testing the overlapping treatment-cell line combinations between ALMANAC and O'Neil, as expected, all the drug combination synergy scores showed significant drops of replicability (0.2 Pearson's r for S, 0.12 for Bliss, 0.18 for HSA, 0.25 for Loewe, and 0.09 for ZIP), while the CSS score still maintained a higher correlation (0.342 Pearson's r). The higher reproducibility of the CSS score, both within and across the studies, suggested that drug combination sensitivity is more reproducible than synergy, which may justify why most of the clinically approved drug combinations rely on their combinatorial efficacy rather than synergy[31,32].

The above result highlights the challenges of predicting cross-dataset drug combinations including (1) the scarcity of overlapped compounds and cell lines between studies, and (2) the variability in the assay and experimental settings, such as the total number and ranges of doses. To combat these challenges, we propose a machine learning model using the following features (Fig. 1c): (1) for both

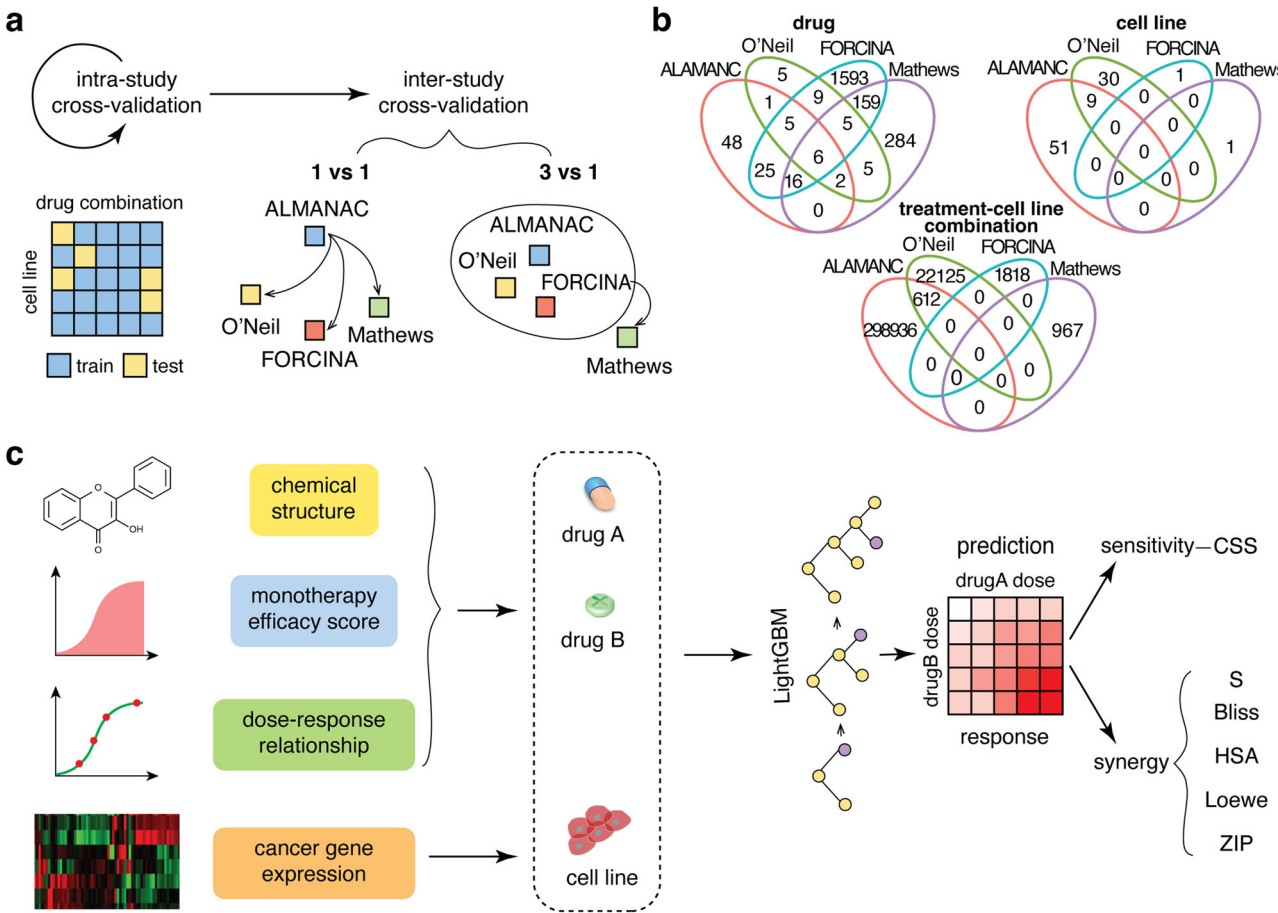

**Fig. 1 Overview of the framework on intra- and inter-study drug combination predictions. a** The cross-validation strategy. We carry out the cross-validation in two steps: intra-study, which is five-fold cross-validation carried out within a single dataset, where the training and test sets are split by drug combination and cell lines, and inter-study, which is carried out between different datasets. The models used in the 1 vs. 1 inter-study cross-validations are the models generated from the inter-study training step. For the 3 vs. 1 inter-study cross-validation, three of the four datasets are combined and used as the training set to generate five models by five-fold cross-validation and then tested on the remaining dataset. **b** The overlapped information (drug, cell line, and treatment-cell line combination) between the four datasets used in this study. **c** The schematic of model construction in this study. We use four different data sources to generate the machine learning model used in this study. For drug-related features, we used chemical structure, monotherapy efficacy score, and their corresponding dose–response relationship. For the treated cancer cell lines, we used the transcription levels of 293 cancer-related genes. The constructed features are input into a lightGBM learner to generate models predicting the six different response metrics of the combination treatment: CSS, which is the sensitivity score representing the efficacy of the combination, and five synergy scores (S, Bliss, HSA, Loewe, and ZIP) representing the degree of interaction between the two drugs.

drugs, we use chemical structure-derived fingerprints, which can be transferred to chemicals that may not be present in the training set; (2) we use pharmacodynamic properties, such as monotherapy efficacy scores and dose–response curves of the drugs. The dose–response curves will be normalized; 3) we use the expression of 273 essential cancer genes[33] to represent the molecular states of the cell lines. The above features will be fed into a lightGBM boosting model, as it has shown higher efficiency than other tree-based algorithms such as XGboost and Random Forest when training on large datasets[34]. We will evaluate the accuracy of predicting the six types of drug combination response scores (i.e. CSS, S, Bliss, HSA, Loewe, and ZIP).

**Combating inter-study variability by integrating monotherapy efficacy and imputation of dose–response curves.** We observe that experimental settings differ not only between different studies, but also within the same study (Supplementary Table 1 and Supplementary Fig. 2). For example, the dose–response matrix ranges from $2 \times 2$ (FORCINA) to $10 \times 10$ (Mathews), and within the O'Neil dataset, both $4 \times 4$ and $4 \times 6$ dose–response matrices

are used. Meanwhile, the dose ranges differ significantly within and between studies (Supplementary Table 1). For example, within the ALMANAC study, more than 40 different doses have been used (Supplementary Fig. 2), and the maximum doses tested for each drug are different due to their distinctive pharmacodynamic properties[4]. Therefore, we precalculate the replicability of monotherapy efficacy scores, in terms of IC$_{50}$, RI, and the distribution statistics (maximum, minimum, mean, and median of all inhibitions in the dose–response curves) within and between different datasets (Supplementary Fig. 3). We notice that RI and IC$_{50}$ show comparable reproducibility within datasets, with Pearson's r of RI ranging from 0.363 (within Mathews) to 1 (within O'Neil), while Pearson's r of IC$_{50}$ ranges from 0.537 (within ALMANAC) to 1 (within O'Neil). However, the replicability of IC$_{50}$ is much lower than that of RI in the cross-dataset analysis, (Pearson's r = 0.084 for IC$_{50}$ versus r = 0.451 for RI between ALMANAC and O'Neil). Most dose–response curve shape statistics show Pearson's r better than or comparable with IC$_{50}$ and RI, either within or between studies, suggesting potential in cross-study prediction (Supplementary Fig. 3).

We start exploring the drug combination response prediction based on the monotherapy responses such as efficacy and dose–response curves (M1–M12, Fig. 2, Supplementary Figs. 4–10, and Supplementary Table 2–5). Three types of features based on monotherapy responses are constructed, denoted as "monotherapy_efficacy", "drc_baseline", and "drc_imputation". "monotherapy_efficacy" is summarized score of the curve ($IC_{50}$ or RI) using

each of the two drugs independently on the treated cell lines, and has often been used in previous benchmark models in drug combination prediction challenges[11,35,36]. The other two features, "drc_baseline", and "drc_imputation", are based on the exact dose–response relationships (Fig. 2a). The dose–response curve baseline model (M1) using the "drc_baseline" feature method is equivalent to the method previously described by Zheng et. al. [8],

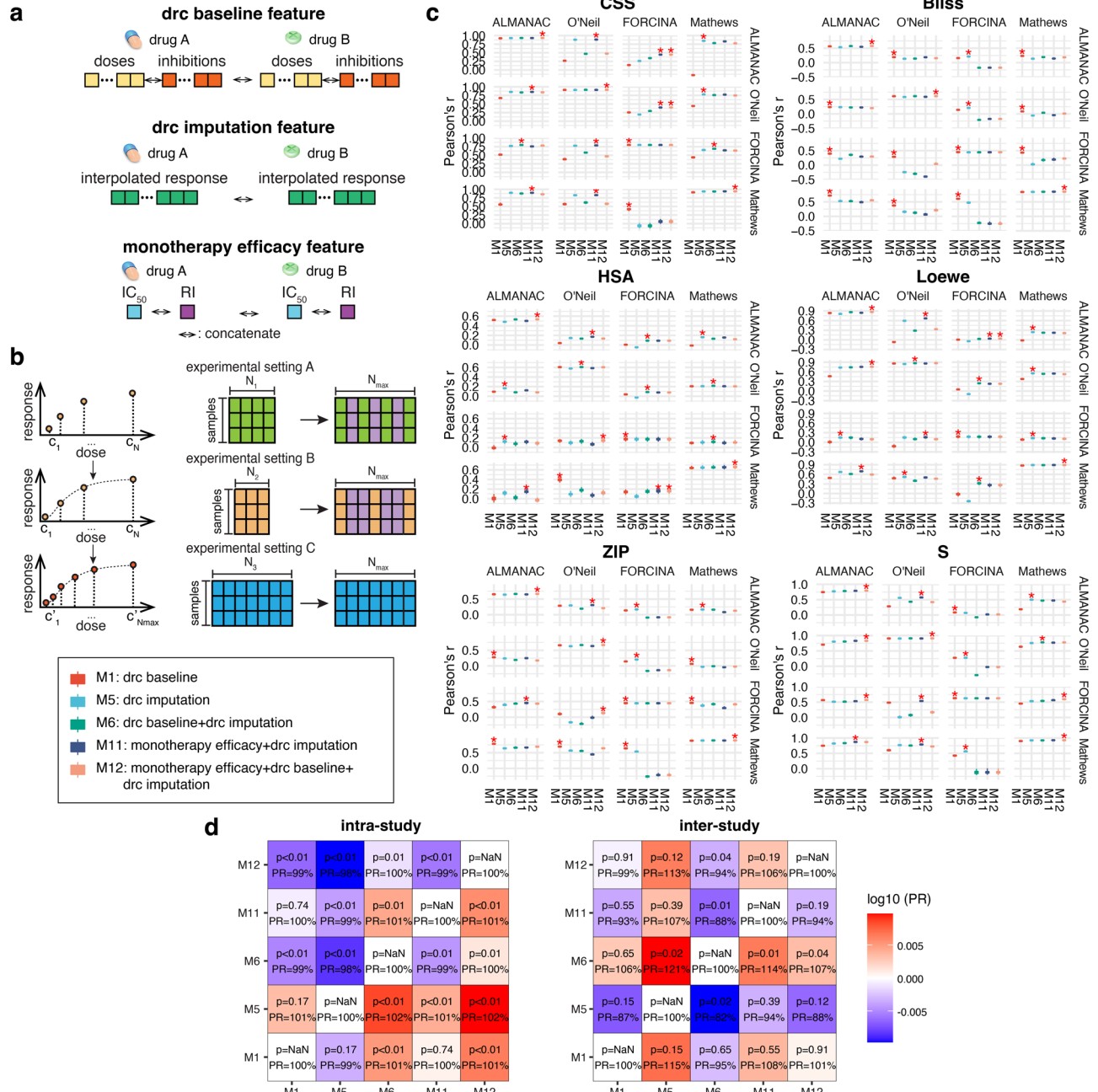

**Fig. 2 Strategy to normalize the differences in inter-study experimental settings. a** Demonstration of different dose–response curves (drc) feature construction schemes. The drc baseline feature is defined as the direct concatenation of doses and corresponding responses, where the total number of doses and responses will be padded by "-1" to the same length for different experimental settings. The drc imputation feature is the concatenation of imputed responses by different interpolation methods (see Methods for details). The monotherapy efficacy feature is the $IC_{50}$ and RI of both drugs on the same cell line. **b** Schematics of inter-study interpolation normalization in experimental settings. For experimental settings A, B, and C, which are tested using a different total number of doses, $N_1$, $N_2$, and $N_3$, we pull out the largest number of doses across all the studies, denoted as $N_{max}$. Then, the dose–response information of each setting is interpolated to the same size as $N_{max}$. **c** Performances are evaluated by Pearson's r for all models, which are models with different combinations of the three features. The top performance in each training set (top) and testing set (right) is denoted by "*". **d** Heatmap shows the results from paired *t* test between the performances of five models in intra- and inter-study cross-validation. The color in the heatmap shows the performance ratios (PRs) of the average performances between each model pair.

where the doses and corresponding responses are vectorized and concatenated directly together. Since the total number of doses varies significantly between different datasets, the "drc_baseline" features were padded to the same length. For the "drc_imputation" feature, we interpolate all the dose–response curves to the same length for all the datasets (Fig. 2b). We test linear, Lagrange, 4-parameter log-logistic regression (LL4) interpolation (M2–M4, Supplementary Figs. 5 and 6). Among the three interpolation methods, linear interpolation performs the best in the intra-study cross-validation while LL4 performs the best in the inter-study cross-validation. Furthermore, combining all three methods shows better performances in both scenarios, and thus is used in the final "imputation" model (M5, Supplementary Fig. 5b, d). Also, since using $IC_{50}$ and RI together are generally better than them alone in the intra- and inter-study cross-validations, the final monotherapy efficacy feature contains both measurements (M7–M9, Supplementary Figs. 7, 8). Five models using different combinations of the monotherapy response-based features mentioned above, are shown in Fig. 2. We notice that M12, which is a combination of all three types of monotherapy features, performs slightly better in the intra-study cross-validation (101–102% performance ratio compared to the other models), while M5, which is the pure imputation model, performs the best in the inter-study cross-validations (107–15% performance ratio compared to the other models), and this advantage is especially significant in the prediction of Bliss (112–113% performance ratio compared to the other models) and Loewe scores (119–138% performance ratio compared to the other models) (Supplementary Fig. 4). It is expected that M12 performs the best in the intra-study validation since the un-imputed dose–response baseline features contain the doses for dose–response evaluation. These doses chosen for monotherapy response evaluation can be significantly different (Supplementary Table 2), thus causing biases in the cross-study prediction. However, the monotherapy doses can still be effective for within-study prediction since it contains unique experimental information for each drug. The imputation method, on the other hand, indeed alleviates the biases in the experimental settings and is more universally transferable between different experimental settings, thus M5, which only imputes dose–response information, outperforms all other monotherapy-based models.

When comparing the monotherapy efficacy directly with dose–response curve-based models, interestingly, the efficacy model shows the best performance in inter-study prediction while the worst in intra-study prediction (Supplementary Figs. 9, 10). We notice that the efficacy model performs especially well when trained or tested on the FORCINA dataset, which adopts a $2 \times 2$ dose–response matrix design (Supplementary Fig. 9a). We reckon that the coarse dose–response relationship may not be as good as the total efficacy in this case, as the imputation becomes unreliable with only two doses.

**The imputation methods improve the benchmarks model's performance in the cross-study prediction.** Previously, the DrugComb study provided a state-of-the-art model using the O'Neil dataset, by integrating one-hot encoding of drugs and cell lines as well as drug chemical fingerprints, drug doses, and cell line gene expressions in the model construction[8]. In this study, we construct a benchmark model based on their schemes, by encoding the chemical structure properties and molecular profiles of drugs and cell lines in the feature set, and explore if the imputation method of the dose–response curve can further improve the prediction accuracy across different individual datasets (Fig. 3 and Supplementary Fig. 11).

We construct five models step-by-step, from the label information (categorical encoding of both drugs and cell lines), to adding the chemical structure of both drugs encoded by molecular fingerprints and cell line cancer gene expression, to adding monotherapy efficacy, and adding the dose–response curve baseline feature and imputation feature, respectively. The performances of all models are listed as M13–M20 (Supplementary Tables 2–5). And five models, including M13-16, and M20, are listed for the main comparison (Fig. 3a).

We notice that, while the benchmark models with only information directly from drugs and cell lines (M13 and M14) still achieve decent performances around the experimental reproducibility levels in intra-study cross-validation (Supplementary Table 2), neither of these models achieve better-than-random performances in the cross-study predictions, due to a lack of shared drugs and cell lines across different studies (Fig. 3b and Supplementary Table 3). Incorporating pharmacological properties such as monotherapy activity on the same cell lines (M15) improves both the intra-study and inter-study prediction performances to 178% and 1299% compared to the reference model (M13), showing the robustness of monotherapy efficacy information between studies (Fig. 3c). Adding the monotherapy baseline information (M6) further improves the inter-study performance but not the intra-study, possibly due to the same reason we mention in the previous section, that the baseline information contains the dose settings, which is a dataset-exclusive artifact. Furthermore, adding the imputed information (M20) further improves the performances in both intra- and inter-study cross-validation, to 184% in the intra-study cross-validation and 1367% in the inter-study validation (Fig. 3c). This improvement is consistent in terms of all the drug combination sensitivity and synergy scores, with 1187% in CSS, 2141% in Bliss, 949% in HSA, 2257% in Loewe, 723% in ZIP, and 2019% in S score, respectively (Supplementary Fig. 11b). Notably the models achieve better performances than experimental replicates within and between studies (Supplementary Tables 2–5). We conclude that the imputed dose–response curve contains orthogonal information to the monotherapy efficacy, which can be effectively used to improve the prediction of combination treatment response by overcoming the variability between different experimental settings.

To understand which information plays the most important role in the inter-study prediction, we carry out SHAP (SHapley Additive exPlanations) analysis to visualize the contribution of all the features in the best-performing model (M20, Fig. 3). As expected, the dose–response curve derived feature shows significant SHAP importance and remains the top feature for all the drug combination response score predictions, while the monotherapy efficacy score also shows significant importance in the S score prediction (Supplementary Fig. 12). We then analyze the contributions of the dose–response imputation features specifically and noticed that the imputed responses at the beginning and end of the curve show significant importance in the prediction, suggesting that the minimum and the maximum response of the monotherapies are informative for predicting the drug combination response (Supplementary Fig. 13).

To demonstrate the robustness of our models in broader inter-dataset validation settings, we carry out 3 vs. 1 cross-validation experiments based on the four datasets we use in this study (Fig. 4 and Supplementary Figs. 14–17). For each training and test setting, we combine three datasets and use the combination as the training set, then test the model on the remaining datasets. We expect that using a multi-sourced training set can lead to improved model performances, by including more types of drugs and cell lines in the training instances. Thus, the training datasets can potentially contain more transferable information to new datasets. As expected, the optimal model in 1 vs. 1 inter-study cross-validation settings, M20, which is the baseline model plus dose–response curve

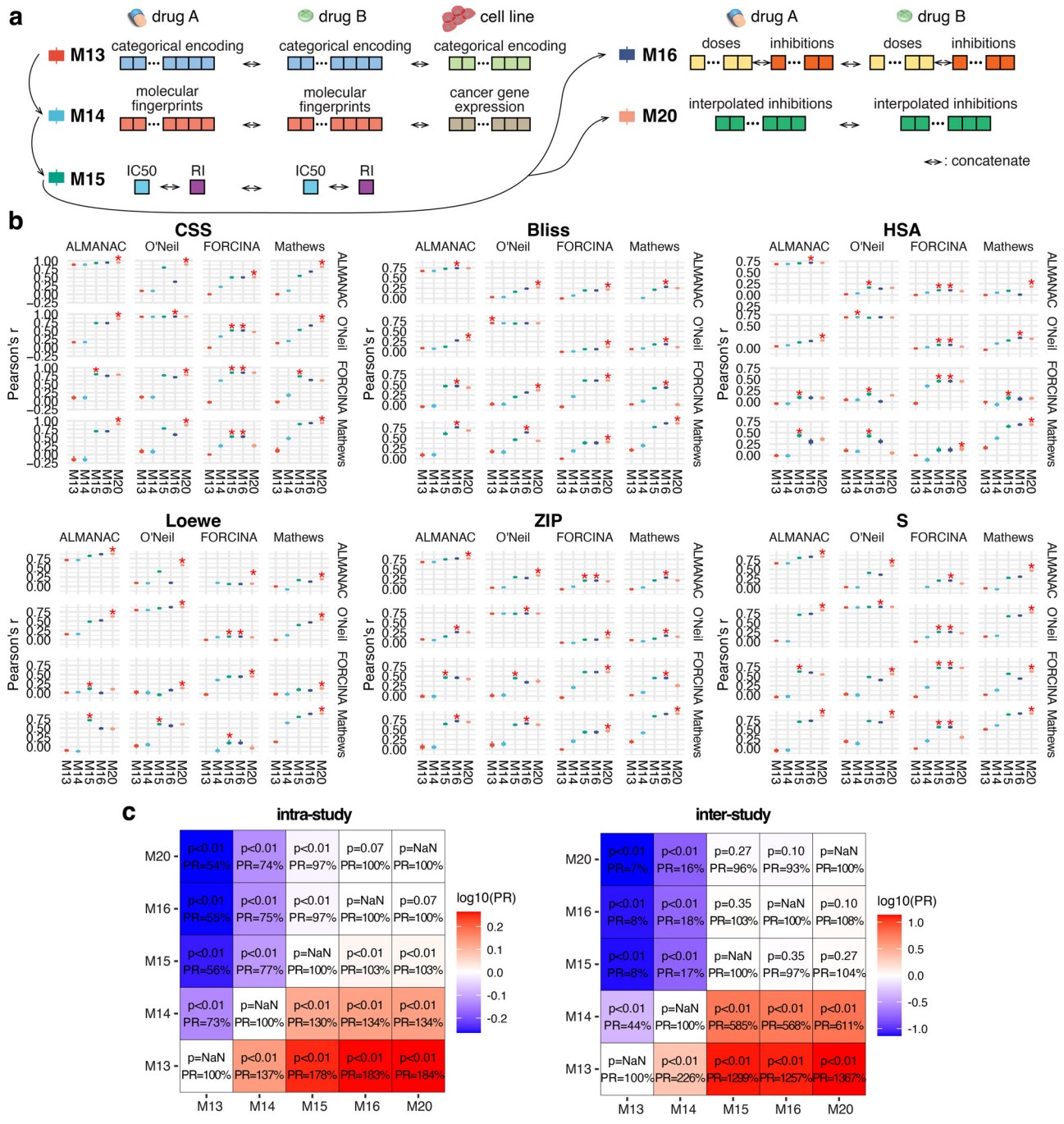

**Fig. 3 Normalized dose–response information improves the intra- and inter-study prediction performances of benchmark models. a** Schematic of the step-by-step feature construction strategy from the benchmark models (M13–M15) to the dose-response-curve-incorporated models (M16 and M20). **b** Performances in all the training and testing scenarios for M1–M5. The best-performing models were denoted by "*" **c** Comparisons of performances of M1–M5 from paired *t* test. The performance ratios (PRs) between model pairs are shown in different colors.

imputation feature, shows the same advantages compared to the other models, with 910% performance compared to M1 and 1544% performance compared to M2 (Fig. 4b).

## Discussion

How to tackle the replicability in results between different studies to draw meaningful conclusions has been a critical issue in drug discovery[37]. During cancer treatment, resistance is frequently developed against monotherapies, and a combination usage of multiple drugs targeting parallel pathways is needed to overcome

this issue. While the application of high-throughput screening on cancer cells accelerates the rational design of drug combinations toward clinical trials[35,38,39], the inconsistency between currently available datasets has been a major concern, posing a challenge to translate these in vitro studies into an in vivo setting[20,40–43]. As the experimental replicability between independent combination screening datasets can be quite low (0.089–0.342 Pearson's *r* between ALMANAC and O'Neil) (Supplementary Fig. 1), which is much lower than that for monotherapy screening (0.194–0.683 $R^2$)[20], a robust machine learning strategy is urgently needed for meaningful clinical applications.

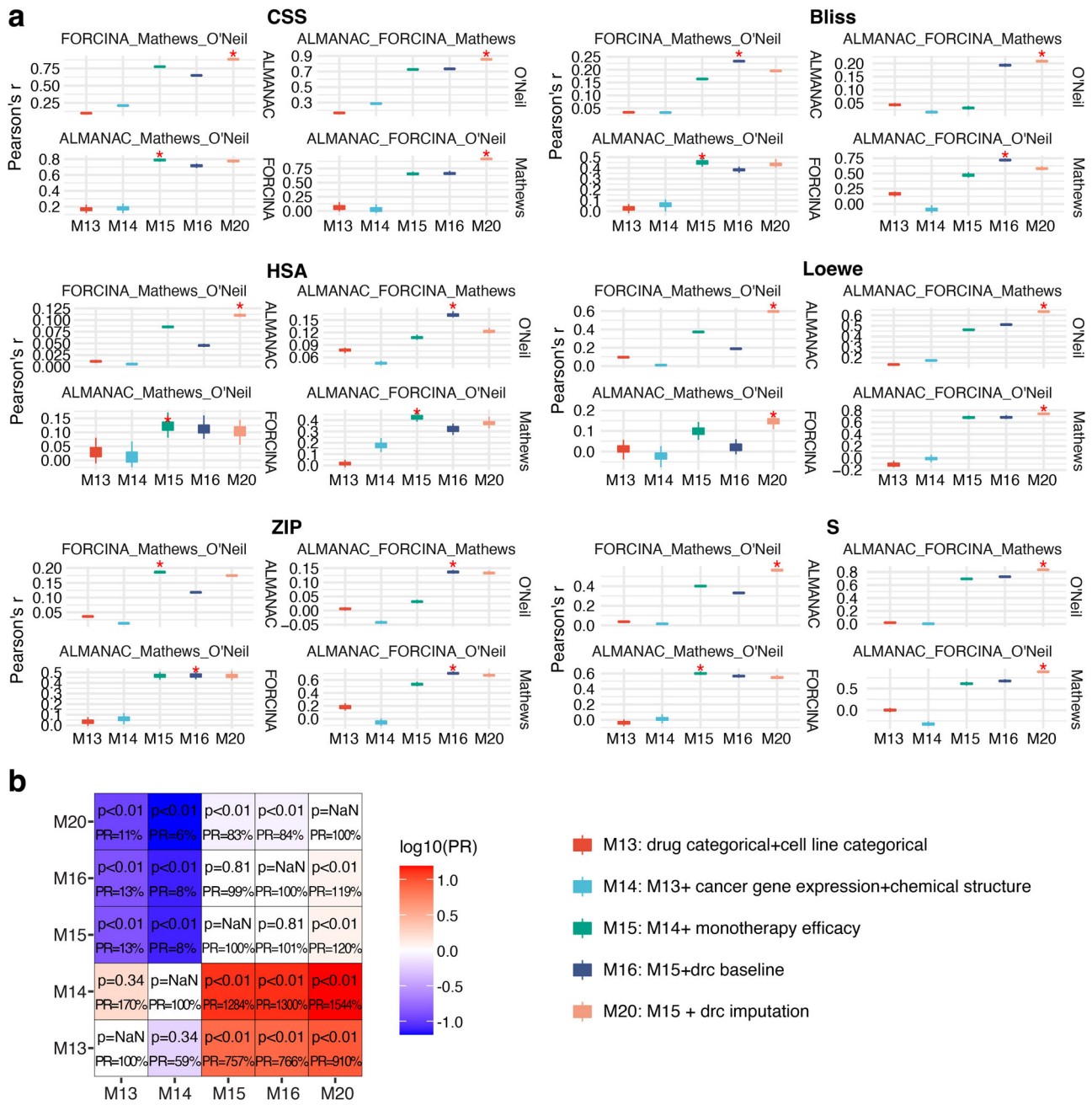

**Fig. 4 Comparison of performances before and after incorporating dose–response curve into the baseline model in inter-study predictions.** Models were trained using three datasets and then tested on the remaining dataset. The models refer to the same model definition in Fig. 3. **a** Performances of machine learning models in the 3 vs 1 training-testing settings. For each comparison, the training set includes three studies shown on the top, while the test set contains one study shown on the right. Top performances are marked by "*". **b** The Pairwise comparison of performances of five models, showing the performance ratios (PRs) and their p-values (paired t test).

Our study addresses the inter-study transferability issue in large-scale screening. We identify a major cause of variability between different studies, which is the experimental setting of drug dosage. The total number of doses, and the dose ranges, can be significantly different between studies, and even between replicates within single studies (Supplementary Fig. 3). Based on the above observation, we consider the dose–response relationship as part of the features in our machine learning model for drug combination sensitivity and synergy prediction and find out that such a modeling strategy significantly improves the transferability of machine learning models between datasets, with an accuracy that is comparable with in-study replicabilities (Supplementary Tables 2–5).

Our study focuses on the transfer learning between in vitro high-throughput drug combination screening studies[44], however, future work is needed to further improve the clinical translation of drug combination predictions. For example, it remains unknown whether the top drug combinations from the in vitro studies are transferable to clinical treatment[32], and whether the response of monotherapy treatment can help infer clinically efficacious combinations[45,46] Furthermore, a mechanistic model on signaling pathways is needed to validate that the predicted drug combination biomarkers can be used for patient stratification in clinical trials[47,48]. Future modeling of transferability should be carried out between in vitro and preclinical studies,

such as patient-derived ex-vivo and mouse models, as well as multiple clinical trial meta-analyses[44,49].

## Methods

**Data collection**. Currently, DrugComb has been the largest public data portal for in vitro high-throughput combination treatment screening studies. We selected the four largest datasets (ALMANAC, O'Neil, FORCINA, and Mathews) from Drug-Comb (https://drugcomb.org/) for the inter and cross-study analysis in this paper, where the detailed comparisons for the four datasets are shown in Supplementary Table 1.

DrugComb provides six metrics (CSS, S, Bliss, HSA, ZIP, Loewe) for the responses of combination treatments, and two metrics ($IC_{50}$ and RI (relative inhibition)) for the response of single drug treatments. The details of the formula of these metrics have been described in Zheng et al.[8]. Briefly, CSS analyzes the overall drug efficacy for the combination treatment, while S, Bliss, HSA, ZIP, and Loewe evaluate the synergy or the degree of interaction between the two drugs used in a combination treatment. Besides the efficacy and synergy metrics for monotherapy/combination therapy, DrugComb also provides the SMILES (Simplified molecular input line entry system) format chemical structure of drugs, which is used for structural encoding in this study.

The transcriptomic profiles of all the cancer cell lines used in this study were obtained from CCLE (Cancer Cell Line Encyclopedia) (https://sites.broadinstitute.org/ccle/datasets). We obtained 279 cancer-associated genes from the IMPACT (Integrated Mutation Profiling of Actionable Cancer Targets) project[33], 273 of which were found to be overlapped with the CCLE transcriptomic profiles. Therefore, these 273 genes were used for combination treatment response prediction in this study.

**Hyperparameters of machine learning models**. We chose the lightGBM gradient boosting model as the base learner used in the experiment. The hyperparameters of the lightGBM models were set as follows:

$$param = \{'boosting\_type' : 'gbdt',$$
$$'objective' : 'regression',$$
$$'num\_leaves' : 20,$$
$$'max\_depth' : 8,$$
$$'force\_col\_wise' : 'true',$$
$$'learning\_rate' : 0.05,$$
$$'verbose' : 0,$$
$$'n\_estimators' : 1000,$$
$$'reg\_alpha' : 2.0, \}$$

where the total number of leaves was set to 20 and the maximum depth was set to 8 to avoid overfitting on the training dataset. 'num_boost_round' was set as 500 for boosting iterations.

**Training and cross-validation of models within and between studies**. For cross-validation of the models, we carried out model training in the following steps:

1. *intra-study training and cross-validation*: in this step, we carried out five-fold cross-validation for model training and testing. We split the training dataset by combination treatment-cell line, therefore the model can be tested on unseen examples to predict new combination treatment synergy and efficacy. As a result, for each of the four datasets, five models were generated by training on different combination treatment-cell line splits. Since the two drugs in the combination should be considered equally, during the training steps, the first and second drugs were switched and put in the training set again to adjust for the possible bias by order of the two drugs.
2. *1 vs. 1 inter-study validation*: in this step, no extra models need to be trained. The models trained within each study from step (1) were used for prediction in other datasets except for the training dataset. In this step, the final prediction results from the five intra-study models generated from step (1) are ensembled by averaging. The ensemble method can reduce the prediction variance thus improving the stability of inter-study prediction performance[50].
3. *3 vs. 1 inter-study validation*: To explore the generalization of *1 vs. 1 inter-study validation* in step (2), we tested the same feature settings on datasets with different compositions. In this step, we combined 3 of the 4 datasets as the training set and tested it on the remaining dataset. The training process is still carried out by inter-study five-fold cross-validation as step (1) and tested on the remaining dataset as step (2).

**Feature preprocessing and construction**. On the input data from the DrugComb data portal (an example of input data is shown in Supplementary Table 6), we applied the following types of information to generate an inter-study-transferable model. The chemical and pharmacological properties of both drugs and the biological characteristics of the treated cell lines were used to construct the feature space.

Firstly, we defined a reference model by applying the following types of information:

1. Categorical encoding of the names of both chemical agents in the treatment (denoted as "drug_categorical"), and categorical encoding of the cancer cell line (denoted as "cell_line_categorical"). Both features were implemented as categorical features during the training of lightGBM models.
2. To provide information in terms of the drugs' chemical properties, we generate molecular fingerprints from the chemical structure of both chemical agents (denoted as "chemical_structure"). 166 MACCS, 1024 Morgan, and 2048 RDK molecular fingerprints were generated based on the SMILES format of the chemical structure of drugs, using *openbabel* and *rdkit* modules from *Python*. The three types of fingerprints were concatenated directly for the chemical structure encoding.
3. To provide a meaningful biological background of the treated cell lines, we used the gene expression levels of 273 cancer-associated genes obtained from CCLE as the representation of the cell line features (denoted as "cancer_gene_expression"). The gene expression levels for each cell line were quantile normalized before implementation.
4. To provide pharmacological properties of the single drugs, we used two efficacy metrics of each of the cancer drugs on the same cell line: $IC_{50}$ (denoted as "monotherapy_ic50") and RI (denoted as "monotherapy_ri"), where $IC_{50}$ represents the dose of the drug achieving 50% of the maximum response, and RI is the normalized area under the log10-transformed dose–response curve.
5. For more detailed pharmacological properties, and also to evaluate the variability of experimental settings in different studies, we used the information from the dose–response dose of the single drugs on the same cell lines, which is also provided by the DrugComb datasets. We encoded the dose–response curves using different methods as follows:

a. dose–response curve baseline encoding (denoted as "drc_baseline"): the doses of and corresponding responses were flattened as a vector and concatenated together. Since in different experiments, the total number of doses measured could be different, ranging from two to ten, the total number of doses is padded to ten by -1 from the right. For example, for the monotherapy MK-5108 tested on the ES2 cell line, the response was measured at five different doses (μm): [0, 0.075, 0.225, 0.675, 2], and the corresponding response is [0, -0.48, -0.47,4.32, 20.72], then both doses and responses will be padded to [0,0.075,0.225,0.675,2,-1,-1,-1,-1,-1] and [0,-0.48, -0.47,4.32, 20.72,-1,-1,-1,-1,-1], and concatenated together for feature input.
b. dose–response curve imputation encoding (denoted as "drc_imputation"): instead of directly taking dose–response curve information as the baseline encoding, we normalized the dose-response relationship by interpolation since the dose-response curves within and between different studies are measured by significantly different dose numbers and ranges (Fig. S4), the total number of responses on the curve can be different, introducing a significant challenge for applying this information in inter-study validation. Therefore, interpolating the dose–response curves to the same length can help them to be interpreted at the same magnitude. While all dose–response relationships were measured at logarithmic dose scales, the maximum length of the dose–response curve ranges from 2-10. Therefore, all dose-response curves are first log10-transformed and then interpolated to the length of 10. We carried out the following commonly used interpolation methods and tested the difference between them:

i. Linear interpolation (denoted as "drc_intp_linear"): we use the *Numpy* Python package to generate the linear interpolated dose–response curve. The linear interpolation is computed using the Eq. (1):

$$y = y_0 + (x - x_0)\frac{y_1 - y_0}{x_1 - x_0} \tag{1}$$

where $(x, y)$ is the coordinate for the interpolated point between $(x_0, y_0)$ and $(x_1, y_1)$.

ii. Lagrange interpolation (denoted as "drc_intp_lagrange"): we used the *Scipy* Python package to compute the Lagrange interpolation of the dose–response curve. The formula for computing Lagrange interpolation is Eq. (2):

$$y = P(x) = \sum_{j=1}^{n} P_j(x) \tag{2}$$

Where,

$$P_j(x) = y_j \prod_{k=1, k \neq j}^{n} \frac{x - x_k}{x_j - x_k}$$

$n$: total number of doses before interpolation.

iii. Four-parameter log-logistic (LL4) regression interpolation (denoted as "drc_intp_4PL"): As dose–response curves are often fitted by a four-

parameter logistic regression function in the standard analysis, we implemented a Python version of the *drc* R package using the same parameter implementation[51]. The LL4 interpolated curve is computed by Eq. (3):

$$y = b + \frac{c - b}{(1 + \exp(a(\log(x) - \log(IC50))))} \tag{3}$$

where,

$$
\begin{aligned}
a &= \frac{y_n - y_1}{x_n - x_1}, \\
b &= y_{\max}, \\
c &= y_{\min}, \\
d &= IC50.
\end{aligned}
$$

In total, 20 different combinations of the above features are tested in this paper. For details on all the models, please refer to Supplementary Table 5, and the corresponding performances are summarized in Supplementary Tables 2-4.

**Visualization of feature importance in machine learning models**. To visualize the feature importance during cross-study validation, we carried out SHAP (SHapley Additive exPlanations) analysis, a game-theory-based AI visualization method, on both individual features and grouped features, by taking the advantage of the addictive nature of Shapley values[52,53]. The SHAP analysis is carried out and plots are generated by using the python *shap* package[54].

**Statistical quantification and evaluation metrics**. The model's performances, as well as the replicability of drug response measurements, are evaluated by Pearson's correlation coefficient (*r*). Pearson's correlation coefficient is defined by Eq. (4):

$$r = \frac{\sum(x_i - \bar{x})(y_i - \bar{y})}{\sqrt{\sum(x_i - \bar{x})^2 \sum(y_i - \bar{y})^2}} \tag{4}$$

Where x is the gold standard and y is the prediction value when evaluating the machine learning model performances. When evaluating the intra- and inter-study experimental replicability, we selected all possible paired permutations from the replicate experiments with the same treatment-cell line combinations and computed the Pearson's r between the two replicates in these permutations. This step demonstrates the variability of experiments and provides a reference for the upper bound for the machine learning model prediction.

As the distribution of each dataset deviates significantly we didn't use RMSE as the main evaluation metric in this study. Since RMSE can be significantly decreased by approaching the average values of all responses, but not as sensitive by distinguishing higher and lower responses in the test dataset. Thus, the models failed to generate meaningful predictions to differentiate combination experiments with different responses that can have lower RMSE. This drawback can be overcome by using a relativity-based metric, such as Pearson's correlation coefficient, instead.

The confidence of evaluation metrics, of the 95% confidence interval, is generated by bootstrapping the predictions from the total datasets. We randomly sampled the prediction results from the test set without replacement 100 times to generate the 95% confidence interval.

Since all models were tested in different training and testing dataset combinations, to evaluate the consistency of model performances in the intra- and inter-study cross-validation, we carried out two-sided paired *t* tests to evaluate the significance of differences between each pair of models, which are calculated as

$$t = \frac{\sum d}{\sqrt{\frac{n(\sum d^2) - (\sum d)^2}{n-1}}} \tag{5}$$

Where d is the difference between each pair and n is the sample size.and are defined as below:

We also use the performance ratio (PR) to compare the average performances of two different models:

$$PR = \frac{\overline{x_1}}{\overline{x_2}} \tag{6}$$

Where $\underline{x_1}$ is the average performance of the first model and $\underline{x_2}$ is the average performance of the second model.

Both performance ratio (PR) and significance of the p-value of the paired *t* test were used to show the magnitude of differences between the two models.

**Reporting summary**. Further information on research design is available in the Nature Portfolio Reporting Summary linked to this article.

## Data availability

The data analyzed in this study can be freely downloaded from DrugComb data portal: https://drugcomb.fimm.fi/.

## Code availability

The source code of the analysis and models are available on GitHub: https://github.com/GuanLab/DrugComb-cross-study-prediction.

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

## Acknowledgements
Y.G. received research support from NIH R35GM133346. JT received research support from the Academy of Finland (No. 317680), and BZ received research support from the Otto A. Malm Foundation and the University of Helsinki Integrative Life Science Doctoral Programme scholarship. J.T. and B.Z. both received research support from the European Research Council (No. 716063).

## Author contributions
Study design: Y.G., J.T. Data analytics: Y.N., H.Z., Z.W., B.Z. Manuscript writing: H.Z., B.Z., J.T., YG. Figures: H.Z., Y.D., Y.N. All authors read and approved the manuscript.

## Competing interests
The authors declare no competing interests.

## Inclusion and Ethics Statement
The data analyzed in this study were obtained from publicly available sources and did not involve any new recruitment of participants, therefore do not require a statement on inclusion/exclusion criteria. The authors affirm that they have complied with the principles of the Declaration of Helsinki throughout the conduct of this study.
