## [Peer Review File · Communications Biology]

Reviewers' comments:

Reviewer #1 (Remarks to the Author):

The strategies described in this article are useful for a scientific community in the fields of drug design and use. Machine-learning approach seems interesting and useful. I see the major weakness of the article in the description of methods. I prefer to see the clear step-by-step explanation of the strategy. Authors starts from the data from several trials, what they do with these data first, next step, etc. The fragments of the input files would be appropriate to show.

Reviewer #2 (Remarks to the Author):

This paper addresses an important problem of inter-study transferability in large-scale drug combination screening, and the authors propose a method to overcome experimental variability by harmonizing dose-response curves across studies. The overall design of the model is neat, and its validity has been demonstrated through extensive experiments.

I have a few concerns that I would invite the authors to address in their revised version:

Major comments:

1. The authors' test results focus on the exploration of different combinations of features based on machine learning models. However, the drug combination problem has been extensively studied in recent years and I think it is important to employ the previous state-of-the-art approach as a benchmark for comparison.

2. To my knowledge, dose-response curves have been applied in previous prediction methods, and although I understand the motivation for using them is different, it would be helpful to the reader if the authors could be more specific in comparing the differences in the methods involved.

3. We can find a significant performance enhancement when compared the proposed method to the baseline model on page 7: "This enhancement is consistent across all drug combination sensitivities and synergies. The scores for sensitivity and synergy were 1187% for CSS and 2141% for HSA. This improvement was consistent across sensitivity and synergy with 1187% for CSS, 2141% for Bliss, 949% for HSA and 2257% for Loewe. Loewe, 723% for ZIP, and 2019% for S score, respectively (Supplementary Figure 11b)". In light of this, I feel that it would be helpful to the reader if the authors could further analyse the effect of the dose-response curve on the predicted outcome through a case study.

Minor comments:

The description before and after the text should be consistent. Is it Fig. Sx or Supplementary Fig. x? Is it x% ~ x% or x%~x%? Also, some grammatical errors should be fixed, e.g. on page 5: "Among the three interpolation methods, linear interpolation performs the best in the intra-study cross-validation while LL4 performs the best in the intra-study cross-validation.". I suggest that the authors double-check the manuscript.

Reviewer #3 (Remarks to the Author):

The manuscript analyses the performances of drug combination prediction models, for different descriptors and label types, and for different experimental dataset and combinations thereof. The authors provide extensive description of the methodologies and of the outcomes for the many different models, analysing the inter- and intra-study variations.

The manuscript presents quite a lot of data, which the authors might provide in a better and clearer

way, for example to provide all the performance plots (Fig 2c, 3b, 4a) with the y-axis (Pearson's r) at the same scale, to allow direct comparison. Similarly, the heatmaps (Fig 2d, 3c, 4b) contain too much information (p and FC), and the authors might explain the FC values how it is calculated, if FC=100% means same performance or performance changed by 100%, and why FC=100% has sometimes $p=0.07$, 0.20 or 0.81, but otherwise NaN.

All comments are marked in black.

All replies are marked in blue.

Reviewers' comments:

Reviewer #1 (Remarks to the Author):

The strategies described in this article are useful for a scientific community in the fields of drug design and use. Machine-learning approach seems interesting and useful. I see the major weakness of the article in the description of methods. I prefer to see the clear step-by-step explanation of the strategy. Authors start from the data from several trials, what they do with these data first, next step, etc. The fragments of the input files would be appropriate to show.

We thank the reviewer for this suggestion. While we appreciate your comments, we believe that the methods described in the article are indeed in a step-by-step manner and that the article provides a clear explanation of the strategy used.

First of all, the data was collected thru the DrugComb portal (<https://drugcomb.org/>), which is also developed by the collaborators in this paper. The input data files are exactly the same as those presented DrugComb. We added a fragment of the input files in the supplementary:

“

Supplementary Table 6. Example of the drug combination dataset provided by DrugComb. For each combination, two drugs (drug_row and drug_col) and the treated cell line (cell_line_name) is shown. Five synergy scores (ZIP, Bliss, Loewe, HSA and S) and sensitivity score (css) are shown for each experiment. For each drug combination, there could be more than one replicated experiment. And the source of each experiment can be traced using the block id.

block id	drug_row	drug_col	cell_line_name	CSS	ZIP	Bliss	Loewe	HSA	S
1	5-FU	ABT-888	A2058	30.869	3.865	6.256	-2.951	5.537	19.839
2	5-FU	ABT-888	A2058	27.46	8.247	12.334	3.126	11.614	16.43
3	5-FU	ABT-888	A2058	29.901	6.063	11.660	2.452	10.941	18.871
4	5-FU	ABT-888	A2058	24.016	-4.280	5.145	-4.063	4.426	12.986

”

The step-by-step strategy of cross-validation settings in this paper has been described in the manuscript as below:

“For cross-validation of the models, we carried out model training in the following steps:

- 1) *intra-study training and cross-validation*: in this step, we carried out five-fold cross-validation for model training and testing. We split the training dataset by combination treatment-cell line, therefore the model can be tested on unseen examples to predict new combination treatment synergy and efficacy. As a result, for each of the four datasets, five models were generated by training on different combination treatment-cell line splits. Since the two drugs in the combination should be considered equally, during the training steps, the first and second drugs were switched and put in the training set again to adjust for the possible bias by order of the two drugs.
- 2) *I vs. I inter-study validation*: in this step, no extra models need to be trained. The models trained within each study from step (1) were used for prediction in other datasets except for the training dataset. In this step, the final prediction results from the five intra-study models generated from step

- (1) are ensemble by averaging. The ensemble method can reduce the prediction variance thus improving the stability of inter-study prediction performance⁴⁹.
- 3) *3 vs. 1 inter-study validation*: To explore the generalization of *1 vs. 1 inter-study validation* in step (2), we tested the same feature settings on datasets with different compositions. In this step, we combined 3 of the 4 datasets as the training set and tested it on the remaining dataset. The training process is still carried out by inter-study five-fold cross-validation as step (1) and tested on the remaining dataset as step (2).”(Page 6)

The detailed feature construction process has been described in the Method section as below:

Feature preprocessing and construction

On the input data from the DrugComb data portal (an example of input data is shown in **Supplementary Table 6**), we applied the following types of information to generate an inter-study-transferable model. The chemical and pharmacological properties of both drugs and the biological characteristics of the treated cell lines were used to construct the feature space.

Firstly, we defined a reference model by applying the following types of information:

- 1) Categorical encoding of the names of both chemical agents in the treatment (denoted as “drug_categorical”), and categorical encoding of the cancer cell line (denoted as “cell_line_categorical”). Both features were implemented as categorical features during the training of lightGBM models.
- 2) To provide information in terms of the drugs’ chemical properties, we generate molecular fingerprints from the chemical structure of both chemical agents (denoted as “chemical_structure”). 166 MACCS, 1024 Morgan, and 2048 RDK molecular fingerprints were generated based on the SMILES format of the chemical structure of drugs, using *openbabel* and *rdkit* modules from *Python*. The three types of fingerprints were concatenated together directly for the chemical structure encoding.
- 3) To provide a meaningful biological background of the treated cell lines, we used the gene expression levels of 273 cancer-associated genes obtained from CCLE as the representation of the cell line features (denoted as “cancer_gene_expression”). The gene expression levels for each cell line were quantile normalized before implementation.
- 4) To provide pharmacological properties of the single drugs, we used two efficacy metrics of each of the cancer drugs on the same cell line: IC₅₀ (denoted as “monotherapy_ic50”) and RI (denoted as “monotherapy_ri”), where IC₅₀ represents the dose of the drug achieving 50% of the maximum response, and RI is the normalized area under the log10-transformed dose-response curve.
- 5) For more detailed pharmacological properties, and also to evaluate the variability of experimental settings in different studies, we used the information from the dose-response dose of the single drugs on the same cell lines, which is also provided by the DrugComb datasets. We encoded the dose-response curves using different methods as follows:
 - a) dose-response curve baseline encoding (denoted as “drc_baseline”): the doses of and corresponding responses were flattened as a vector and concatenated together. Since in different experiments, the total number of doses measured could be different, ranging from two to ten, the total number of doses is padded to ten by -1 from the right. For example, for the monotherapy MK-5108 tested on the ES2 cell line, the response was measured at five different doses (μm): [0, 0.075, 0.225, 0.675, 2], and the corresponding response is [0, -0.48, -0.47, 4.32, 20.72], then both doses and responses will be padded to [0,0.075,0.225,0.675,2,-1,-1,-1,-1,-1] and [0,-0.48, -0.47,4.32, 20.72,-1,-1,-1,-1,-1], and concatenated together for feature input.
 - b) dose-response curve imputation encoding (denoted as “drc_imputation”): instead of directly taking dose-response curve information as the baseline encoding, we normalized the dose-response relationship by interpolation since the dose-response curves within and between different studies are measured by significantly different dose numbers and ranges (Fig. S4), the

total number of responses on the curve can be different, introducing a significant challenge for applying this information in inter-study validation. Therefore, interpolating the dose-response curves to the same length can help them to be interpreted at the same magnitude. While all dose-response relationships were measured at logarithmic dose scales, the maximum length of the dose-response curve ranges from 2-10. Therefore, all dose-response curves are first log10-transformed and then interpolated to the length of 10. We carried out the following commonly-used interpolation methods and tested the difference between them:

- i) Linear interpolation (denoted as “drc_intp_linear”): we use the *Numpy* Python package to generate the linear interpolated dose-response curve. The linear interpolation is computed

$$y = y_0 + (x - x_0) \frac{y_1 - y_0}{x_1 - x_0}$$

using the equation (1):

... .. (1)

where (x, y) is the coordinate for the interpolated point between (x_0, y_0) and (x_1, y_1) .

- ii) Lagrange interpolation (denoted as “drc_intp_lagrange”): we used the *Scipy* Python package to compute the Lagrange interpolation of the dose-response curve. The formula for computing Lagrange interpolation is equation (2):

$$y = P(x) = \sum_{j=1}^n P_j(x)$$

... .. (2)

Where,

$$P_j(x) = y_j \prod_{k=1, k \neq j}^n \frac{x - x_k}{x_j - x_k}$$

n : total number of doses before interpolation.

- iii) Four-parameter log-logistic (LL4) regression interpolation (denoted as “drc_intp_4PL”): As dose-response curves are often fitted by a four-parameter logistic regression function in the standard analysis, we implemented a Python version of the *drc* R package using the same parameter implementation⁵⁰. The LL4 interpolated curve is computed by equation (3):

$$y = b + \frac{c - b}{(1 + \exp(a(\log(x) - \log(IC50))))}$$

... .. (3)

where,

$$a = \frac{y_n - y_1}{x_n - x_1},$$

$$b = y_{max},$$

$$c = y_{min},$$

$$d = IC50.$$

In total, 20 different combinations of the above features are tested in this paper. For details on all the models, please refer to **Supplementary Table 5**, and the corresponding performances are summarized in **Supplementary Table 2-4**.

”(Page 7-8)

Reviewer #2 (Remarks to the Author):

This paper addresses an important problem of inter-study transferability in large-scale drug combination screening, and the authors propose a method to overcome experimental variability by harmonizing dose-response curves across studies. The overall design of the model is neat, and its validity has been demonstrated through extensive experiments.

I have a few concerns that I would invite the authors to address in their revised version:

Major comments:

1. The authors' test results focus on the exploration of different combinations of features based on machine learning models. However, the drug combination problem has been extensively studied in recent years and I think it is important to employ the previous state-of-the-art approach as a benchmark for comparison.

We thank you for this suggestion.

It's true that the drug combination prediction problem has been extensively studied before and a dozen of methods have been applied (Menden et al. 2019; Güvenç Paltun et al. 2021). Our team (Yuanfang Guan team) was the top performer from the previous AstraZeneca DREAM challenge in drug combination prediction (Menden et al. 2019). Our collaborator also published DrugComb (Zagidullin et al. 2019; Zheng et al. 2021), which is one of the most up-to-date repository and analytic portals for large-scale high-throughput drug combination screening datasets.

In our paper, the benchmark model on page 4 in the Results section: “**The imputation methods improve the benchmarks model’s performance in the cross-study prediction**” is the state-of-the-art approach adapted from the DrugComb analytic portal (Zheng et al. 2021).

To address this question, the following explanations is added in the Results section:

“Previously, the DrugComb study provided a state-of-the-art model using the O’Neil dataset, by integrating one-hot encoding of drugs and cell lines as well as drug chemical fingerprints, drug doses, and cell line gene expressions in the model construction 8. In this study, we construct a benchmark model based on their schemes, by encoding the chemical structure properties and molecular profiles of drugs and cell lines in the feature set, and explore if the imputation method of the dose-response curve can further improve the prediction accuracy across different individual datasets (**Fig. 3** and **Supplementary Fig. 11**)” (Page 4)

2. To my knowledge, dose-response curves have been applied in previous prediction methods, and although I understand the motivation for using them is different, it would be helpful to the reader if the authors could be more specific in comparing the differences in the methods involved.

We thank the reviewer for this suggestion.

It is true that dose-response curves have been applied in previous prediction methods, such as by Ianevski et al. (Ianevski et al. 2019), who predicts the responses on different dose levels instead of the whole dose-response matrices. They filled out the incomplete dose-response matrix, which is often used in real-life experiments, to the full matrix and found that a sparse dose-response matrix can provide enough information. Wang et al. developed comboFM and comboLTR, which is similar to Ianevski et al, fills the sparse dose-response matrices by imputation and predicts the full dose-response matrices of the new drug combinations (Wang et al. 2021). Zheng et al., on the other hand, implemented only the concentration information, including the concentrations and the maximum concentration in their model by one-hot encoding (Zheng et al. 2021).

However, for all the methods we mentioned above, their model implementation is only applicable when training and predicting on datasets the same dose-response matrix settings. That is, the different lengths of dose-response matrices will result in different sizes in input features, therefore, cross-dataset comparison using models that utilize dose-response matrices has never been applicable to this challenging situation before. Whether or not their implementation can maintain the same performances between different datasets has never been discussed.

In our paper, we select the method from DrugComb paper (Zheng et al. 2021), which is the state-of-the-art method in predicting combination prediction, as the benchmark model. In this paper, the model achieved 0.98 Pearson's correlation in predicting inhibition, 0.79 correlation in predicting HSA, 0.89 correlation in predicting ZIP, 0.57 in predicting Loewe, and 0.87 in predicting Bliss, on the O'Neil dataset.

The DrugComb implemented the dose-response matrices as one-hot encoding of the maximum doses, and also the vectorized dose-response matrix. To adapt this method between different dose ranges, we pad all vectorized dose-response matrices to the same length (10X10). This is what the **drc_baseline** model referred to in M1.

To provide a more specific explanation of the different implementations of dose-response models in our paper, we added the following paragraph to the Results part:

“We start exploring the drug combination response prediction based on the monotherapy responses such as efficacy and dose-response curves (M1-M12, **Fig. 2, Supplementary Fig. 4-10, and Supplementary Table 2-5**). Three types of features based on monotherapy responses are constructed, denoted as “monotherapy_efficacy”, “drc_baseline”, and “drc_imputation”. “monotherapy_efficacy” is summarized score of the curve (IC_{50} or RI) using each of the two drugs independently on the treated cell lines, and has often been used in previous benchmark models in drug combination prediction challenges^{11,35,36}. The other two features, “drc_baseline”, and “drc_imputation”, are based on the exact dose-response relationships (**Fig. 2a**). The construction of the dose-response curve baseline model (M1) using the “drc_baseline” feature, is based on the method previously described by Zheng et. al.⁸, where the doses and corresponding responses are concatenated directly together. Since the total number of doses varies significantly between different datasets, the “drc_baseline” features were padded to the same length. For the “drc_imputation” feature, we interpolate all the dose-response curves to the same length for all the datasets (**Fig. 2b**). We test linear, Lagrange, 4-parameter log-logistic regression (LL4) interpolation (M2-M4, **Supplementary Fig. 5 and 6**).” (Page 3-4).

3. We can find a significant performance enhancement when compared the proposed method to the baseline model on page 7: "This enhancement is consistent across all drug combination sensitivities and synergies. The scores for sensitivity and synergy were 1187% for CSS and 2141% for HSA. This improvement was consistent across sensitivity and synergy with 1187% for CSS, 2141% for Bliss, 949% for HSA and 2257% for Loewe. Loewe, 723% for ZIP, and 2019% for S score, respectively (Supplementary Figure 11b)". In light of this, I feel that it would be helpful to the reader if the authors could further analyse the effect of the dose-response curve on the predicted outcome through a case study.

We thank the reviewer for this great suggestion. A case study could a great suggestion to analyze the effect of the dose-response curve on the predicted outcome.

However, as our model is based on a large dataset with around half a million dataset, using SHAP analysis, to visualize the decision-making process of our final model, rather than just one case study, should be a much better way to demonstrate the importance of dose-response curve based features. We have conducted SHAP analysis in our results section as follows:

Figure S12. Feature contribution of best performing model (M20 in Figure 3) in inter-study prediction when trained on ALMANAC and tested on O’Neil study. The importance when predicting all six response scores were shown below.

This shows the dose-response-curve-based features (“drc_imputation”), have significantly larger contributions than other features. To describe this effect, we add the following paragraph in the Results

section: “To understand which information plays the most important role in the inter-study prediction, we carry out SHAP (SHapley Additive exPlanations) analysis to visualize the contribution of all the features in the best-performing model (M20, **Fig. 3**). As expected, the dose-response curve derived feature shows significant SHAP importance and remains the top feature for all the drug combination response score predictions, while the monotherapy efficacy score also shows significant importance in the S score prediction (**Supplementary Fig. 12**). We then analyze the contributions of the dose-response imputation features specifically and noticed that the imputed responses at the beginning and end of the curve show significant importance in the prediction, suggesting that the minimum and the maximum response of the monotherapies are informative for predicting the drug combination response (**Supplementary Fig. 13**).”(page 5)

Minor comments:

The description before and after the text should be consistent. Is it Fig. Sx or Supplementary Fig. x? Is it x% ~ x% or x%~x%? Also, some grammatical errors should be fixed, e.g. on page 5: "Among the three interpolation methods, linear interpolation performs the best in the intra-study cross-validation while LL4 performs the best in the intra-study cross-validation.". I suggest that the authors double-check the manuscript.

We thank the reviewer for pointing out these formatting errors.

We have corrected all supplementary figures citations in the manuscript to Supplementary Fig. x (marked as blue in original manuscript.)

Also, we corrected all x% ~ x% to x%~x% (marked as blue in original manuscript).

For the sentence on Page 5, we corrected the sentence as: "Among the three interpolation methods, linear interpolation performs the best in the intra-study cross-validation while LL4 performs the best in the inter-study cross-validation."

Reviewer #3 (Remarks to the Author):

The manuscript analyses the performances of drug combination prediction models, for different descriptors and label types, and for different experimental dataset and combinations thereof. The authors provide extensive description of the methodologies and of the outcomes for the many different models, analysing the inter- and intra-study variations.

The manuscript presents quite a lot of data, which the authors might provide in a better and clearer way, for example to provide all the performance plots (Fig 2c, 3b, 4a) with the y-axis (Pearson's r) at the same scale, to allow direct comparison. Similarly, the heatmaps (Fig 2d, 3c, 4b) contain too much information (p and FC), and the authors might explain the FC values how it is calculated, if FC=100% means same performance or performance changed by 100%, and why FC=100% has sometimes p=0.07, 0.20 or 0.81, but otherwise NaN.

We thank the reviewer for this suggestion. It is a good idea to present the data in a better and clearer way. However, As there are 20 different models, and each section in this paper are discussing different questions, providing all performance plots at the same scale may not be a good idea.

For example, Fig. 2d focuses on discussing the sole usage of concentrations. Comparing using solely concentrations, with the full model, which definitely has much better results since it incorporates more different types of information, will blur out the differences between different concentration models in Figure 2, impede the clarity of our discussion and impact our whole manuscript structure. The usage of the concentration model is our main novelty in this paper, and it's the first time we found that concentration can be used to solve the disparity between different datasets. Therefore, the contents of Figure 2 are worth to be taken out as a single section.

At the same time, the models in Figure 4 can not be put in the same figure as Figure 2 and Figure 3. Since they are not trained and tested on the same datasets (Figure 4 uses three datasets for training, while Figure 2 and Figure 2 use one dataset for training), therefore, they are not on the same magnitude scale.

We also thank the reviewer's suggestion on clearing the information in the heatmaps. As we have mentioned in our figure legends, the heat map is the results from the paired t-test for model performances against each other, and the FC is the fold change of average performances between each model pair. The statistics of paired test is calculated as below:

$$t = \frac{\sum d}{\sqrt{\frac{n(\sum d^2) - (\sum d)^2}{n-1}}}$$

Where d is the difference between each pair and n is the sample size. In the heatmaps, p=NAN is because the the model is compared with itself, therefore the difference d is constantly 0. Therefore t, in this case, is Nan.

As the FC is only the fold change of the average value, the exact value of the two samples can be different even tho they have the same mean. Therefore, the t statistics of the paired t-test do not necessarily relate to each other. This is also why it is necessary to present both FC and p values at the same time.

We understand that some readers without statistical background may not be as familiar with paired t-test. Therefore, we added the formula of paired t-test to the method (page 8-9):

“Since all models were tested in different training and testing dataset combinations, to evaluate the consistency of model performances in the intra- and inter-study cross-validation, we carried out two-sided paired t-tests to evaluate the significance of differences between each pair of models, which are calculated as

$$t = \frac{\sum d}{\sqrt{\frac{n(\sum d^2) - (\sum d)^2}{n-1}}} \dots \dots (5)$$

Where d is the difference between each pair and n is the sample size. and are defined as below:

We also use the fold change (FC) to compare the average performances of two different models:

$$FC = \frac{\overline{x}_1}{\overline{x}_2} \dots \dots (6)$$

Where \overline{x}_1 is the average performance of the first model and \overline{x}_2 is the average performance of the second model.

Both fold-change (FC) and significance of the p-value of the paired t-test were used to show the magnitude of differences between the two models.”

REVIEWERS' COMMENTS:

Reviewer #2 (Remarks to the Author):

The authors have improved the paper according to my previous comments and suggestions well. Thus, I recommend to accept this paper.

Reviewer #3 (Remarks to the Author):

The manuscript has answered all/most reviewer's comments in a satisfactory way.

One remaining issue is the misleading terminology used for FC as Fold Change. The provided formula defines it actually as a Performance Ratio ($\text{Ratio} = X1/X2$), with a value=1 representing the same performance value for model1 and model2.

100% fold change implies a 100% difference in performance between model1 and model2, with the appropriate formula for a change(!) to be $\text{FC} = (X1 - X2)/X1$ or $/X2$..

FC should be renamed as something like Performance Ratio (PR), to satisfy the reader with statistical background.

Comments are marked in black

Replies are marked in Blue.

REVIEWERS' COMMENTS:

Reviewer #2 (Remarks to the Author):

The authors have improved the paper according to my previous comments and suggestions well. Thus, I recommend you accept this paper.

We thank the reviewer for his recommendation.

Reviewer #3 (Remarks to the Author):

The manuscript has answered all/most reviewer's comments in a satisfactory way.

One remaining issue is the misleading terminology used for FC as Fold Change. The provided formula defines it actually as a Performance Ratio ($\text{Ratio} = X1/X2$), with a value=1 representing the same performance value for model1 and model2.

100% fold change implies a 100% difference in performance between model1 and model2, with the appropriate formula for a change(!) to be $\text{FC} = (X1 - X2)/X1$ or $/X2$..

FC should be renamed as something like Performance Ratio (PR), to satisfy the reader with statistical background.

We thank the author for this suggestion.

Usually in bioinformatics, a fold-change also means fold-increase or decrease, mathematically defined as: $\text{FC} = X2/X1$. For example, in specific bioinformatics tasks such as differential gene expression analysis, the fold change has always been used to compare the the gene expression between two groups of samples and the \log_2 fold change has been used to draw a volcano plot, which is usually symmetric at $\log_2(\text{FC})=0$, where $\text{FC}=1$ means no change.

However, the fold-change also has an alternative definition, as the reviewers have mentioned, mathematically defined as $\text{FC} = (X1 - X2)/X1$. We understand the confusion of the reviewer and decided to make the following adjustments.

Due to the disparity between these two definitions, we decided using a Performance Ratio (PR) rather than Fold Change (FC) may be more appropriate in this case. We changed all Fold Change (FC) in text and figures to Performance Ratio (PR) in our manuscript.